# The Role of Electroencephalogram-Assessed Bandwidth Power in Response to Hypnotic Analgesia

**DOI:** 10.3390/brainsci14060557

**Published:** 2024-05-30

**Authors:** Mark P. Jensen, Tyler D. Barrett

**Affiliations:** Department of Rehabilitation Medicine, University of Washington, Seattle, WA 98195, USA; barre386@uw.edu

**Keywords:** hypnosis, hypnotic analgesia, pain electroencephalogram, EEG, theta, alpha

## Abstract

Research supports the efficacy of therapeutic hypnosis for reducing acute and chronic pain. However, little is known about the mechanisms underlying these effects. This paper provides a review of the evidence regarding the role that electroencephalogram-assessed bandwidth power has in identifying who might benefit the most from hypnotic analgesia and how these effects occur. Findings are discussed in terms of the slow wave hypothesis, which posits that brain activity in slower bandwidths (e.g., theta and alpha) can facilitate hypnosis responsivity. Although the extant research is limited by small sample sizes, the findings from this research are generally consistent with the slow wave hypothesis. More research, including and especially studies with larger sample sizes, is needed to confirm these preliminary positive findings.

## 1. Introduction

Research supports the efficacy of clinical hypnosis for reducing both acute and chronic pain [1,2]. Although these findings are important for making decisions regarding treatment, the findings from clinical trials that focus only on the effects of hypnosis on clinical outcomes do not help us understand the mechanisms that underlie hypnosis’s effects on pain. Thus, although we know that clinical hypnosis can reduce pain, we do not yet know *how* hypnosis has these effects, and we have very limited knowledge regarding *for whom* hypnosis is most beneficial.

Both theory and preliminary research suggest the possibility that brain activity—specifically, brain activity as measured by electroencephalography (EEG)—may both underlie the effects of hypnosis on pain (i.e., serve as a mediator of hypnotic analgesia) and predict who is most likely to benefit from hypnotic analgesia (i.e., serve as a moderator of hypnotic analgesia) [3,4]. This review summarizes the extant evidence regarding the role of EEG-assessed bandwidth power as a possible mediator and moderator of response to hypnotic analgesia. It begins with a description of the concepts of mediation and moderation and summarizes the importance of identifying the factors that mediate and moderate treatment outcome. The next section describes what EEG measures and discusses the possible role that activity in different bandwidths has in understanding brain states and an individual’s phenomenological experience. Next, the role of the magnitude of different EEG-assessed bandwidths in understanding hypnosis is described, and the slow wave hypothesis is introduced. The primary section of the article follows, in which the research that has been conducted to examine the slow wave hypothesis is reviewed. A discussion of the clinical and research implications of the findings follows.

### 1.1. Treatment Mediators and Moderators

The “How does hypnosis work?” question can be addressed by research that identifies the *mediators* of treatment outcome [5,6]. A variable is deemed a treatment mediator if analyses show that the beneficial effects of treatment are explained, at least in part, via its effects on that variable. For example, a number of theories argue that psychological pain treatments are effective because they reduce pain-related catastrophizing, increase a patient’s sense of control over pain, or result in changes in other pain-related beliefs [7]. In this case, pain-related beliefs are considered to be potential treatment mediators. This idea can be tested using data from clinical trials that include measures of the potential mediator(s) at multiple outcome assessment points. Analyses can then be conducted that test the effects of treatment on the mediators and the subsequent effects of the mediators on outcome [8].

The “For whom does the treatment work?” question is usually addressed with research that identifies the *predictors* and *moderators* of treatment outcome. A variable is identified as a predictor if it is assessed before treatment begins and analyses show that this variable is associated with treatment outcome for the treatment(s) that is (are) subsequently provided. A variable is identified as a treatment moderator if it is significantly associated with a response to one treatment more than for another treatment [5,6]; predictor variables predict the outcome for all treatments, while moderator variables predict the outcome for some treatments but not others. For example, hypnotizability has sometimes been shown to predict response to hypnosis treatments in both laboratory [1] and clinical settings [2], although the ability of hypnotizability to predict response to hypnotic analgesia in clinical populations tends to be weak [9]. If research results show that hypnotizability predicts response to other treatments in addition to hypnosis treatments, it can be viewed as a general treatment outcome predictor. On the other hand, if hypnotizability is found to predict response to hypnosis treatments more than other pain treatments, it can be labeled as a treatment outcome moderator.

Research to identify treatment mediators, predictors, and moderators has several important theoretical and clinical implications [10]. First, an increased understanding of treatment mechanisms can bring order to the field. If treatments with proven efficacy are found to be effective through their effects on a small subset of key mediators, this knowledge could help to develop a better understanding of how treatments that are thought to be different may actually be quite similar, at least with respect to their underlying mechanisms [11]. Second, by identifying the most important treatment mediators, treatment efficacy could be maximized by adapting treatments to focus more on these mechanism variables than variables that do not drive positive outcomes. For example, if changes in catastrophizing beliefs are found to be more important mediators than changes in pain self-efficacy beliefs, then treatment outcomes could be improved by targeting catastrophizing beliefs instead of self-efficacy beliefs. Finally, an understanding of treatment mechanisms can help identify potential treatment moderators, permitting better matching of patients to treatment. For example, if decreases in a specific pain-related belief are found to be a key mediator, then individuals who are assessed as having high levels of this belief before treatment would be more likely than individuals who are assessed as having low levels of this belief to benefit from a treatment that effectively changes this belief [12].

### 1.2. Electroencephalography-Assessed Bandwidth Power as a Possible Mediator, Predictor, and Moderator of Hypnosis Treatment

#### 1.2.1. What Does Electroencephalography Measure?

Electroencephalography (EEG) measures of activity in different bandwidths have been identified as potential mediators, predictors, and moderators of hypnotic analgesia. EEG measures the electrical activity of the neurons on the surface of the cortex via electrodes placed directly on the scalp. In the brain, groups of these neurons—called neuronal assemblies—fire together at different frequencies, which can range from very slow (i.e., <once/second, or <1 Hz) to very fast (100s of times/second, or >100 Hz). This firing of neuronal assemblies [13] at the same frequency has been hypothesized to be the physiological representation of our experience of a specific thought, experience, image, or idea [14]. According to this model, one neuronal assembly fires when we see or experience a specific color. Another assembly would fire when we see or imagine a beach.

The raw EEG signal represents the activity of neurons that are firing across all frequencies within the sensor range of each electrode placed on the scalp. The magnitude or amplitude of this signal is referred to as *power*. The raw EEG signal can be grouped into different frequency bandwidths, with the most common being labeled delta (e.g., just over 0 to 4 Hz), theta (4 to 8 Hz), alpha (8 to 12 Hz), beta (12 to 30 Hz), and gamma (>30 Hz), although these are sometimes further classified into smaller bandwidths (e.g., low vs. high theta).

One of the most interesting things about bandwidth power is that it is associated with different states or phenomenological experiences [15]. For example, a preponderance of delta power is associated with “diminished” consciousness, such as occurs during deep sleep [16,17]. Relatively high levels of theta power are associated with either (or both) sleepiness or focused attention [18,19]. Relatively high alpha power is associated with relaxed mental states [20,21], as well as “creative ideation” [22]. More beta power is associated with perceived stress [23], and both beta and gamma are positively associated with information processing, including more “thinking” [24,25]. EEG-assessed bandwidth power has also been found to be associated with noxious stimulation that is assumed to contribute to pain, with such stimulation associated with an increase in delta and beta power and decrease in alpha power [26].

#### 1.2.2. EEG-Assessed Bandwidth Power and Hypnosis

Perhaps in part because early models of hypnosis viewed hypnosis as a specific “state” of consciousness distinct from other waking states, and in light of the evidence noted in the previous section that measures of bandwidth power are associated with different states of consciousness, there was an early interest in examining the role of bandwidth power as a potential reflection of hypnosis, or as reflecting the mechanisms underlying the effects of hypnosis (cf. [27,28,29]).

The findings from EEG-based hypnosis research have been summarized by several researchers [3,30,31,32]. A number of conclusions can be drawn from this body of research, including: (1) beta power is not likely to be a predictor or mediator of response to hypnosis treatments; (2) additional research is needed to determine if delta power is a predictor or moderator of response to hypnosis treatments; (3) statistically significant effects do not always emerge between measures of bandwidth power and response to hypnotic suggestions, but when they do, theta is the bandwidth most commonly found to predict response to hypnosis; (4) when significant effects emerge for alpha, more alpha power tends to be associated with more response to hypnosis; and (5) when significant effects emerge for gamma, both hypnosis-related increases and decreases in gamma power have been found.

When considered as a group, the existing research findings have led us to develop what we refer to as the “slow wave” (i.e., mostly theta but sometimes low alpha) hypothesis [3,33]. This hypothesis proposes that (1) greater slower wave power assessed just before or during hypnosis can contribute to an enhanced response to hypnotic suggestions and that (2) neuronal assemblies that fire in slower wave bandwidths operate by controlling neuronal assemblies that fire in faster bandwidths (e.g., gamma) to create the responses to hypnotic suggestions. Another way of stating this is that neuronal assemblies that fire in the theta and alpha frequencies control the activity of neuronal assemblies that fire in gamma frequencies, with activity in the latter underlying what we experience. This idea is based in part on the known role that theta and alpha activity plays in memory recall (e.g., “Imagine yourself on a beach…” or “The sky is so very blue…”) and for recording memories (e.g., “…the benefits you have obtained from this session will stay with you and linger beyond the session…”) [34,35].

The slow wave hypothesis has yet to be confirmed. However, preliminary support for this hypothesis comes from one study that examined differences in hypnotizability as a function of time of day [36]. Using two large databases that contained hypnotizability scores assessed in large samples of undergraduate students, investigators found higher hypnotic responsiveness scores when the measures were administered in the late morning and early evening compared to when they were administered at other times of the day. They also noted that other researchers have found similar patterns when hypnotizability was assessed in the same person at different times [37,38]. Bandwidth power changes in a reliable way over the course of the day, such that most oscillation bandwidths are at their nadir in the early morning, increase during the day, and peak in the afternoon or evening [39]. However, there is one exception to this rule: relative theta power and the low end of alpha (i.e., 8 Hz) peak at two times during the day—late morning and early evening [39]. Thus, the times in the day when theta and low alpha power appear to be relatively high are also the times of day when hypnotizability appears to be higher, consistent with the slow wave hypothesis [40].

#### 1.2.3. Bandwidth Power and Hypnotic Analgesia

Given the findings that, when statistically significant findings emerge, measures of alpha and theta and power are found to be associated with response to hypnotic suggestions in general, it is reasonable to hypothesize that activity in these bandwidths might be associated with response to hypnotic analgesia in particular. We therefore conducted a review of the literature to evaluate the state of evidence regarding this possibility.

## 2. Method

To identify the articles to review, we conducted a search of PubMed on 30 April 2024, of all articles indexed using the keywords *hypnosis* and *EEG* and *pain* and *delta* or *theta* or *alpha* or *beta* or *gamma*. The search was conducted without the use of quotation marks around the keywords to cast the widest net possible. We then read the titles and abstracts of these articles to identify any that examined bandwidth power as a potential predictor, moderator, or mediator of the effects of hypnotic analgesia on pain. We excluded articles that were reviews or that did not focus on pain as an outcome. We also included a recently published article known to the authors that examined baseline EGG-assessed measures as a potential moderator of hypnosis treatment, which was not identified in the PubMed search.

## 3. Results

The initial search yielded 29 articles, 7 of which met the study inclusion and exclusion criteria. With the inclusion of the additional article known to the authors, eight articles were identified for review. The key findings from these articles that are related to the slow wave hypothesis are presented in Table 1.

The articles can be organized with respect to their presentation of findings related to: (1) baseline power in different bandwidths as predictors of response to hypnotic analgesia; (2) the effects of techniques that increase slow wave bandwidth power as a potential strategy for enhancing response to hypnotic analgesia; and (3) the mediation effects of EEG bandwidth power on the effects of hypnotic analgesia. With respect to the first two types of studies, the slow wave hypothesis predicts that more slow wave power assessed before treatment and strategies that increase slow wave power would be associated with greater response to subsequent hypnotic analgesia. With respect to mediation studies, the slow wave hypothesis predicts that the pain reduction effects of hypnotic analgesia mediate the effects of hypnosis by increasing slower bandwidth (theta and alpha) power and decreasing faster bandwidth (beta and gamma) power.

### 3.1. Baseline Power as a Predictor of Response to Hypnotic Analgesia

Three of the eight articles identified examined the extent to which baseline bandwidth power predicts response to hypnotic analgesia. Two of these were pilot studies with low sample sizes. In the first, Freeman and colleagues [41] measured pain intensity in response to a cold pressor task (i.e., placing one’s hand in ice water, which is usually experienced as painful soon after placement), as well as both low theta (3.5–5.5 Hz) and high theta (5.5–7.5 Hz) bandwidth power in 10 highs (i.e., those with high scores on a hypnotizability measure) and 10 lows (i.e., those with low scores on a hypnotizability measure) in three conditions: waking relaxation, distraction, and hypnosis. Consistent with other studies finding that trait hypnotizability predicts response to hypnotic suggestions in laboratory pain studies, they found that the highs reported more pain reduction in the hypnosis condition than in the two control conditions. In fact, they found that the highs reported more pain reduction than the lows across *all* (i.e., not just hypnosis) conditions. They also found that highs evidenced significantly (using an alpha of 0.10 given the low sample size) more high theta power as assessed by electrodes placed over the parietal and occipital areas during both the waking relaxation and hypnosis conditions.

In another study, we examined the ability of bandwidth power assessed just before five different treatment conditions to predict pain reduction following these conditions: (1) an audio recording of hypnotic analgesia, (2) an audio recording of a meditation exercise, (3) neurofeedback to increase alpha power (8–12 Hz) and decrease high beta power (18–30 Hz) over the temporal lobes, (4) transcranial direct stimulation (tDCS) to activate activity in the motor cortex, and (5) sham tDCS [4]. We found that more baseline theta power at 16 of 19 electrode sites (11 of these were statistically significant) was associated with more pain reduction in the hypnosis condition. Baseline theta power did not predict response to any of the other conditions, with the exception of *less* theta power at two posterior sites being significantly associated with response to neurofeedback. The only other significant predictor of response to hypnosis was that having less gamma activity as assessed from two electrodes over the left anterior sites predicted response to hypnosis (i.e., reflecting less information processing in the dorsolateral prefrontal cortex, consistent with the findings that reducing activity in this area using repetitive transcranial magnetic stimulation increases measures of hypnotizability [47]). In summary, despite the low sample size, baseline theta power across the majority of electrodes evidenced an ability to predict response to hypnotic analgesia, and this ability was unique to hypnosis.

Finally, we recently conducted a series of planned secondary analyses to identify possible moderators of the effects of four different psychological treatments for chronic pain using data from a clinical trial that had a large sample (N = 173) [46]. The treatment conditions examined were: (1) four sessions of cognitive therapy (CT), (2) hypnosis focused on pain reduction (HYP), (3) hypnosis focused on changing maladaptive pain-related thoughts (HYP-CT), and (4) a pain education control condition (ED). Measures of EEG-assessed bandwidth power were among the possible moderators examined. Alpha power was identified as a moderator for predicting reductions in pain intensity in response to CT and HYP-CT, with higher levels of baseline alpha power predicting a better response (i.e., more pain reduction) in response to HYP-CT and lower levels of baseline alpha predicting a better response to CT. Delta power was identified as a moderator for predicting reductions in pain intensity in response to HYP-CT, with lower levels of delta power predicting a better response to HYP-CT. Gamma power was also identified as a moderator of pain reductions in response to HYP-CT, with lower levels of baseline gamma power predicting better treatment response. Neither theta nor beta power were identified as significant treatment moderators.

We cannot conclude that the findings from these three studies provide strong confirmatory support for the slow wave hypothesis. However, the findings from the two pilot studies are *consistent with* the hypothesis that theta power is associated with a positive response to hypnotic analgesia, specifically. The findings with respect to alpha power from the larger clinical trial are also consistent with the slow wave hypothesis (but note that a significant effect for theta power did not emerge in this study). As a group, these studies indicate that more research to examine baseline trait theta and alpha power as predictors of response to hypnotic analgesia is warranted.

### 3.2. Enhancing Response to Hypnotic Analgesia by Enhancing Theta Power

Three studies were identified that examined the potential effects of increasing slow wave activity as a strategy for enhancing response to hypnotic analgesia. All three studies combined neurofeedback with hypnosis. With neurofeedback, the individual is provided with feedback, usually in the form of a sound (e.g., music) or image on a computer screen (e.g., a colored bar indicating power in the bandwidth being trained). The feedback provided reflects the level of a pre-identified bandwidth power. The individual is then asked to relax, listen to (or watch) the feedback, and do whatever is needed to change that feedback. As the sound or image changes, so does bandwidth power.

In the first of these studies, Melzack and Perry assigned 24 individuals with chronic pain to one of three conditions: (1) neurofeedback plus an audio recording of hypnosis (12 participants), (2) an audio recording of hypnosis sessions only (6 participants), or (3) neurofeedback alone (6 participants) [42]. The participants in the neurofeedback and hypnosis condition received two sessions of hypnosis, two sessions of alpha neurofeedback with hypnosis, six sessions of alpha neurofeedback, and two sessions during which they were invited to practice the techniques they had learned (i.e., twelve treatment sessions total). The participants in the hypnosis condition received four sessions of listening to the hypnosis audio recording plus two practice sessions (i.e., six treatment sessions total). The participants in the neurofeedback condition received eight sessions of neurofeedback plus two practice sessions (i.e., ten treatment sessions). Outcome was assessed with the McGill Pain Questionnaire, which assesses both pain intensity and pain quality [48]. The investigators found larger improvement (i.e., pain reductions) in both the sensory and affective dimensions of pain in the participants assigned to the combined treatment than in those who received hypnosis only. Moreover, those who received hypnosis only reported larger improvements than those who received neurofeedback only. However, it is important to view these findings as preliminary only, as the small sample size did not provide adequate power to test for statistically significant effects, and the differences in the number of sessions between the conditions may have biased the results in favor of the combined group relative to the other groups.

More recently, we conducted two pilot studies to evaluate the potential for theta power neurofeedback to enhance response to hypnotic analgesia. In the first of these, 20 individuals with multiple sclerosis and chronic pain were given one in-person hypnosis session followed by four pre-recorded hypnosis sessions, with each session providing different suggestions thought to help with pain management [43]. The participants were randomly assigned to receive 20 min of theta neurofeedback training or 20 min of relaxation training just before they listened to the four audio recording sessions. Pain intensity was assessed at pre-treatment, post-treatment, and at 1-month follow-up. Eighteen participants completed all sessions. As the sample size was too small to allow for adequate power to test for statistically significant differences between the conditions, we estimated effect sizes for change in pain from pre-treatment to both post-treatment and 1-month follow-up. Consistent with the slow wave hypothesis, participants in the neurofeedback plus hypnosis condition reported larger effect size improvements in pain intensity from pre-treatment to post-treatment (Cohen’s *d* = 0.70, medium to large effect) and from pre-treatment to follow-up (*d* = 1.04, large effect) than participants in the relaxation plus hypnosis condition (both *d*s = 0.47).

In a second study, we examined the potential hypnosis-enhancing effects of both neurofeedback and training in mindfulness [33], with the rationale that mindfulness training might enhance hypnosis because it is known to increase slow wave power (e.g., [49,50]). In this study, 32 individuals with multiple sclerosis and chronic pain, fatigue, or both were randomly assigned to receive six sessions of theta neurofeedback, six sessions of mindfulness training, or neither. All participants then received one in-person hypnosis session followed by four sessions of audio-recorded hypnosis that included suggestions for greater comfort and energy. Those assigned to the neurofeedback and mindfulness conditions received in-person neurofeedback or mindfulness training just before the last four audio-recording-based hypnosis sessions. Outcomes were assessed at pre-treatment, pre-hypnosis treatment, post-treatment, and 1-month follow-up. Of the participants who reported bothersome pain at pre-treatment, those in the neurofeedback and hypnosis condition reported larger pain reductions from pre-treatment to follow-up (*d* = −1.01, large effect) than those in the mindfulness and hypnosis condition (*d* = −0.36, small effect) or hypnosis only condition (*d* = −0.43, small to medium effect).

As with the two studies examining bandwidth power as a moderator of treatment outcome, the three studies reviewed in this section cannot be used as proof for the slow wave hypotheses. However, the results are generally consistent with the slow wave hypothesis, suggesting that individuals who receive neurofeedback training to increase alpha or theta power will experience greater pain reductions with hypnosis treatment (see [51] for evidence supporting the idea that increasing theta activity can increase hypnotizability in general; a finding also consistent with the slow wave hypothesis).

### 3.3. Bandwidth Power as a Mediator of Hypnotic Analgesia

The search only identified two articles that have been published that have examined the extent to which EEG-assessed bandwidth power may serve as a mediator of the effects of hypnosis on pain. In the first of these, De Pascalis and colleagues studied, among other things, the mediating effects of alpha power on both hypnotic analgesia and a placebo cream on pain intensity [44]. These investigators assessed pain intensity, perceived involuntariness of experiencing hypnotic analgesia, low alpha (alpha1; 7 Hz to 10 Hz), and high alpha (alpha 2; 10 Hz to 12 Hz) during waking and during hypnosis in 65 women in two conditions: (1) aversive cold stimulation and (2) aversive cold stimulation after application of a placebo cream. They found that a higher level of left-temporoparietal alpha2 power (but not alpha1 power) was associated with pain reduction in the hypnosis condition, and that this effect was mediated by perceived involuntariness. Other measures of bandwidth power, including delta, theta, beta, or gamma power, were not assessed.

Finally, in a large (N = 173) randomized clinical trial comparing the effects of four individual sessions of four psychological pain treatments (hypnosis targeting pain reduction, hypnosis targeting changes in pain-related maladaptive thoughts, cognitive therapy targeting changes in pain-related maladaptive thoughts, and pain education), we examined the potential mediation effects of pre- to post-treatment changes in resting state bandwidth power on pre- to post-treatment reductions in pain intensity [45]. We found that *none* of the four treatments had a significant effect on resting state EEG activity for any of the five bandwidths examined (i.e., delta, theta, alpha, beta, and gamma). Moreover, none of the naturally occurring pre- to post-treatment changes in the resting state bandwidth power measures were associated significantly with pain reduction. These findings are inconsistent with the slow-wave-hypothesis-based prediction of resting-state slow bandwidth power assessed before and after the hypnosis sessions as a mediator of hypnotic analgesia. However, this study did not evaluate the effects of hypnosis on bandwidth power assessed *during* hypnosis. It remains possible that slow bandwidth power assessed during the hypnosis session is associated with and/or mediates the effects of hypnotic suggestions on pain reduction experienced. Research is needed to test this hypothesis.

Given the fact that only two studies have been published regarding the mediation effects of slower or faster wave oscillations on hypnotic analgesia, definitive conclusions regarding the slow wave hypothesis based on the available findings cannot be made. However, the findings reported by De Pascalis and colleagues, cited above, regarding the effects of alpha2 as a mediator of the beneficial effects of hypnotic analgesia in response to aversive stimulation in healthy women is consistent with the slow wave hypothesis. Moreover, the lack of any mediation effects of either slower or faster oscillation power during rest (i.e., not during hypnosis) on the beneficial effects of average pain intensity in a fully powered clinical trial suggests the possibility that any mediation, if it exists, might occur during hypnosis and not outside of the hypnotic context. Research is needed to test this possibility.

### 3.4. Summary of the Findings with Respect to the Slow Wave Hypothesis

Overall, the findings from the eight studies identified are generally consistent with the slow wave hypothesis for (1) identifying people who might be more likely to respond to hypnotic analgesia (i.e., those with more theta or alpha power before treatment), (2) identifying strategies that could enhance response to hypnotic analgesia treatments (i.e., interventions or approaches that might increase theta or alpha power), and (3) understanding how hypnosis might reduce pain (i.e., by increasing slower wave power *during hypnosis*). However, there are very few research studies that have been published that test the slow wave hypothesis, and most of those that have been published are pilot studies that were conducted with very few subjects. More research, including more research with larger sample sizes, is needed to determine whether (and under what conditions), the slow wave hypothesis can be confirmed.

## 4. Discussion

### 4.1. Research Implications

The primary research implication of the extant studies is clear: additional research to test the slow wave hypothesis in studies with larger sample sizes is warranted. That said, a caution needs to be made with respect to interpreting research findings in this area. Given the complexities of bandwidth power assessment and the many factors that can impact bandwidth power, it is important to recognize that there is a significant risk for type II errors in EEG-based research. That is, there is a risk for concluding that an effect is not present in the population based on a lack of significant effects in any one study, when in fact the effect may be present in the population.

As one example, theta power can reflect either or both a lack of sleep/sleepiness or focused attention [52]. Higher levels of theta can also be found in some individuals with chronic pain, which has been attributed to a condition called thalamocortical dysrhythmia [53]. Thus, even if the slow wave hypothesis is accurate, and higher levels of slow wave activity can enhance response to hypnosis suggestions during hypnosis (perhaps in this case, the theta associated with positive responding might be labeled “good theta”), there appears to be other slow wave activity that is unrelated to response to hypnosis or hypnotic analgesia suggestions, or that could potentially even interfere with response to hypnotic analgesia (perhaps in this context, this might be labeled “bad theta”). When assessing EEG activity over the entire scalp, the positive and negative impacts of theta may be washed out, resulting in null (i.e., non-significant) findings, even when the “good theta” power that is present, but that cannot be identified due to the presence of “bad theta”, might be having a positive impact. Given this, it would be more important to examine trends in the findings from multiple studies, anticipating that while (1) statistically significant effects will not always emerge, (2) when statistically significant effects do emerge, and if the trend is found for the majority of the significant effects to be consistent with the slow wave hypotheses (as is the case for the studies reviewed in this chapter), the slow wave hypothesis can be considered to remain viable. On the other hand, if approximately 50% of significant findings are consistent with the slow wave hypothesis and 50% of the significant findings are inconsistent with the slow wave hypothesis, this could be used as evidence against support for the slow wave hypothesis.

As alluded to earlier, additional research is needed to examine EEG-assessed bandwidth power as a mediator of response to hypnotic analgesia. In particular, it would be important to examine the effects of hypnotic inductions on alpha and theta power during the induction and while the hypnotic analgesia suggestions are being offered; prior mediation research has focused more on resting state EEG-assessed bandwidth power before and after the hypnosis session or hypnosis treatment. It is possible that bandwidth power *during* hypnosis may play a larger mediation role in response to suggestions than bandwidth power assessed before and after hypnosis.

Finally, more research is needed to evaluate the effects of strategies or interventions that could increase alpha and theta power on response to hypnotic suggestions for pain relief. Such strategies could include, among others, neurofeedback, training in mindfulness meditation practices, or music [33,43,54].

### 4.2. Clinical Implications

If research continues to emerge that is consistent with the slow wave hypothesis, then this research has important clinical implications. First, the findings would suggest that it would be possible to enhance response to hypnosis treatment by interventions that increase slow wave activity. Among the most obvious is to precede hypnotic suggestions with hypnotic inductions, as the latter have been shown to increase theta and alpha power [55,56,57,58,59,60,61]. For those clinicians who have training and experience in neurofeedback, they can consider combining neurofeedback with hypnosis as a strategy for enhancing the beneficial effects of the latter [33,43,62]. It is also possible that some music might be used to enhance theta power, and therefore enhance response to hypnosis [63,64].

A second clinical implication is to be aware that a patient’s theta and alpha power will likely ebb and flow, like all other biological phenomena. As a result, there will be times when patients will be more or less likely to respond to a specific hypnotic suggestion, depending on the level of alpha and/or theta power. Two practical implications of this are to (1) identify the signs that a patient or client is more ready to respond to hypnotic suggestions, and then provide the hypnotic suggestion during a period of greater responsivity, and (2) repeat the most important suggestions a number of times, under the assumption that this will increase the chances that the suggestion will be heard during a time of relatively high responsivity. With respect to the first of these options, clinicians could pay very close attention to specific non-verbal indication of their patient’s state (e.g., muscle tone in the face and/or shoulders, breathing rate) when they offer suggestions, noting which indications are associated with response to suggestions for that patient.

Finally, given the finding reported previously that EEG patterns evidence a specific time pattern over the course of a day, with theta power evidencing a “bump” relative to other bandwidths in the late morning and early evening [36,40], if a clinician is working with a patient who is struggling with responding to hypnosis treatment, they could consider scheduling that patient for a late morning appointment and recommend that the patient listen to their practice audio recordings in the late morning and early evening on a daily basis.

## 5. Summary and Conclusions

In this article, we (1) reviewed the concepts of mediation and moderation and their importance, (2) described the slow wave hypothesis, (3) reviewed research studies that provide empirical evidence regarding the slow wave hypothesis for understanding hypnotic analgesia, and (4) discussed the research and clinical implications of these findings. Although the available evidence is generally consistent with the slow wave hypothesis, much more research is needed to provide enough evidence to draw conclusions regarding its accuracy and utility.

In the meantime, to the extent that the slow wave hypothesis is accurate, clinicians could potentially enhance the efficacy of their hypnosis treatment by: (1) providing treatments or strategies that increase slower wave power prior to hypnosis treatment; (2) observing patients for signs that they may be more likely to respond to hypnotic suggestions, and providing hypnotic suggestions during these times; (3) ensuring that the most important hypnotic suggestions are repeated not only during the session, but also between sessions, by encouraging patients to listen to an audio recording of the suggestions; and (4) scheduling patients who might need extra help for a late morning appointment and/or encouraging patients to listen to audio recordings in the late morning and/or early evening on a daily basis. As more is learned about the mechanisms underlying effective hypnosis treatment, more specific and effective clinical approaches will likely follow. Ultimately, this research and these ideas will result in more individuals benefiting more from hypnosis treatment.

## Figures and Tables

**Table 1 brainsci-14-00557-t001:** Description of included studies on EEG-assessed bandwidth activity and hypnotic analgesia.

Authors (Date); Study Population; Sample Size	Type of Hypnosis	Control Group(s)	Key Findings Related to the Slow Wave Hypothesis
Freeman et al. [41]; healthy sample; N = 20	Audio recording of Stanford Hypnotic Clinical Scale (SHCS) induction followed by suggestions for numbness and “no sensations”.	Two groups: (1) waking relaxation and (2) distraction.	Highly hypnotizable participants evidenced significantly (*p* < 0.10) more high theta power than controls, as measured by electrodes over parietal and occipital areas during hypnosis and waking.
Jensen et al. [4]; spinal cord injury and chronic pain; N = 30	Recording of a countdown induction followed by suggestions for reduced pain and negative pain-related thoughts.	Four groups: (1) meditation, (2) EEG biofeedback, (3) transcranial direct stimulation, and (4) sham transcranial direct stimulation.	Baseline theta power across the majority of electrodes evidenced an ability to predict response to hypnotic analgesia; less gamma activity measured by two electrodes over left anterior sites predicted response to hypnotic analgesia.
Melzack and Perry [42]; chronic pain; N = 24	20 min audio-recorded adaptation of Hartland’s Ego Strengthening Technique followed by suggestions for improving physical, emotional, and psychological function.	Two groups: (1) alpha neurofeedback and (2) hypnosis plus alpha neurofeedback.	All three groups evidenced an increase in alpha power. Those in the hypnosis plus alpha neurofeedback condition evidenced the largest improvements in the sensory and affective dimensions of pain.
Jensen et al. [43]; multiple sclerosis; N = 20	One face-to-face session followed by four 20 min audio-recorded sessions, plus theta neurofeedback (NF-HYP). Each session provided different suggestions thought to help with pain management.	One face-to-face session plus four audio-recorded sessions preceded by audio of ocean waves for 20 min (RLX-HYP).	Participants in the hypnosis plus theta neurofeedback condition reported larger improvements in pain intensity than those in the relaxation plus hypnosis condition.
Jensen et al. [33]; multiple sclerosis and chronic pain and/or fatigue; N = 32	Six sessions of theta neurofeedback training followed by one in-person hypnosis session, followed by four more neurofeedback + audio-recorded hypnosis sessions that included suggestions for relaxation, ego strengthening, improved sleep, more energy, analgesia, and an improved future self.	Two groups: (1) six sessions of training in mindfulness followed by the same five sessions of hypnosis as the theta training group, preceded by mindfulness training + the same audio recorded hypnosis sessions as given to the theta neurofeedback training group; and (2) five sessions of the same hypnosis regimen as the other two groups.	Participants in the theta neurofeedback plus hypnosis condition reported larger pain reductions than participants in either of the other groups.
De Pascalis et al. [44]; healthy sample; N = 65	Two sessions: one to assess hypnotic suggestibility using the Stanford Hypnotic Susceptibility Scale, Form C (Italian Version), and the other using the SHCS induction followed by suggestions for the analgesic effects of a placebo cream.	Waking condition.	During hypnosis, higher levels of left temporoparietal alpha2 predicted pain reduction. This effect was mediated by perceived involuntariness.
Jensen et al. [45]; chronic pain; N = 173	Four in-person sessions of hypnosis targeting either pain reduction (HYP) or changes in pain-related thoughts (HYP-CT).	Two groups: (1) pain education and (2) cognitive therapy.	None of the treatments showed a significant effect on resting EEG activity assessed before and after treatment on the five bandwidths assessed (delta, theta, alpha, beta, and gamma).
Jensen et al. [46]; chronic pain; N = 173	Four in-person sessions of hypnosis targeting either pain reduction (HYP) or changes in pain-related thoughts (HYP-CT).	Two groups: (1) pain education and (2) cognitive therapy.	Participants in the HYP-CT condition who had higher resting state levels of alpha at baseline reported greater improvements in pain intensity. Participants in the CT condition who had lower alpha at baseline reported greater improvements in pain intensity. Participants in the HYP-CT condition who had less delta and gamma power reported greater improvements in pain intensity. Neither baseline theta nor baseline beta power was found to moderate treatment outcome.

Note: SHCS = Stanford Hypnotic Clinical Scale; NF-HYP = theta neurofeedback plus hypnosis; RLX-HYP = relaxation plus hypnosis; HYP = hypnosis targeting pain reduction; HYP-CT = hypnosis targeting pain-related thoughts.

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
