# Peer review of "The Role of Electroencephalogram-Assessed Bandwidth Power in Response to Hypnotic Analgesia"

_brainsci, 2024, doi:10.3390/brainsci14060557_

Round 1

Reviewer 1 Report

Comments and Suggestions for Authors

The author proposes a review about EEG power implication in hypnoanalgesia. This is a nice article and important since EEG findings related on hypnosis are not easy to resume and are often heterogenous. Although the content of the article is very interesting, I think the structure should be re-work slightly. Indeed, we need a more precise description of the method, with clear description of research done by the author. A flowchart should be added to clarify the way the article was constructed on based on some selected articles. And a table could probably be helpful to give to the reader a quick overview of studies related to the main goal of the author. In addition, I have noticed several typos, but I didn’t search for it, so the author should read carefully all the manuscript to check for typos. Here are my detailed comments:

1.     The authors wrote « the findings from outcome research does not help us to understand the mechanisms that underlie hypnosis’s effects on pain. Thus, although we know that clinical hypnosis can reduce pain, (…)”. I do not totally agree with these assumptions. Indeed several studies showed how hypnosis modulates brain activation (EEG, evoked-potential, MRI) and helps to pave the way to the understanding of hypno-analgesia.

2.     Line 33, typos: “tahat”should be corrected

3.     Line 35: “brian” and “individaul's” should be corrected

4.     Line 39 : “implicaitaons” should be corrected

5.     Line 58 “varaibles » should be corrected

6.     Line 59 “moderatir » should be corrected

7.     Line 125 “stasticialy significant” should be corrected

8.     Line 169 “aseline”

9.     Line 191 “hyposis”

10.  Line 290 “lo[46]w »

11.  Line 311 « hyponsis »

12.  A figures illustrating the different EEG rythms could be useful to help the reader to have a visual representation of what it is explained theoretically.

13.  Line 111, the following reference could be added since it also review EEG in hypnosis: Vanhaudenhuyse A, Nyssen AS, Faymonville ME. Recent Insight on How the Neuroscientific Approach Helps Clinicians. OBM Integrative and Complementary Medicine 2020; 5(2): 028; doi:10.21926/obm.icm.2002028

14.  Line 115-116, the author states “ (…) and in light of the evidence, noted in the previous section, that measures of bandwidth power are associated with different states of consciousness,(…)”. However there is no description of different states of consciousness associated with different rhythm. And I think that in the context of this article, a brief description of the association between EEG rhythms and non-ordinary states of consciousness could be interesting. In addition, at some point, the author mentioned more detailed EEG rhythm such as for example alpha-1 and alpha-2. These should explained to the reader, what do these rhythms mean or are related to?

15.  The method should be better describe and a method section should be added to this article. Please add the inclusion/exclusion criteria of articles selected, was there a delay selected (i.e., 10 years before 2024, more?). Did you keep all the 848 articles found on Pubmed? In addition, it seems that the author made a second research based only on pain-related articles, EEG and hypnosis. So the method behind this review is not so clear.

16.  This article is missing a results section summarizing the number of articles found with the keywords, the number that the author keep. A flowchart should be added.

17.  Line  125 “stasticialy significnat effects do not always emerge, but then they do,” à this sentence is not clear for me, could you rephrase? What does the author mean by no statistical results and then they appear?

18.  Lines 122-128: I guess these brief descriptions of the results obtained in EEG studies focused specifically on the effect of hypnosis on pain perception? This should be precise since it’s not clear . In fact this comments relevant for all the section “EEG-Assessed Bandwidth Power and Hypnosis” à does the author presents results related to hypnosis in general or related to pain modulation?

19.  For all studies described in the “results” section, the author should briefly describe the type of hypnosis (which kind of suggestion) and the content of control groups. It could be heavy to add it in the text, so a table resuming the studies selected could be useful for the reader.

20.  Lines 238-245: this sentence is very long to read. Could the author separate it in two sentences?

21.  Line 311 “However, this study did not evaluate the effects of hyponsis on bandwidth power assessed during hypnosis.”. I’m not sure to well understand this conclusive sentence for this study. Could the author clarify his idea here?

22.  The author stated that “If research continues to emerge that is consistent with the slow wave hypothesis, then this research has important clinical implications.” And if this theory is not confirmed by research, what could be the implications? This should be discussed also.

23.  The author proposes to “learn to identify the signs that a patient or client is more ready to respond to hypnotic suggestions, and then provide the hypnotic suggestion during a period of greater responsivity”. Could the author explain how we can learn to identify these signs?

Comments on the Quality of English Language

See my comments

Reviewer 2 Report

Comments and Suggestions for Authors

The review article is well-organized and effectively presents the current state of research on the role of electroencephalogram (EEG)-assessed bandwidth power in response to hypnotic analgesia. The structure of the review facilitates understanding and navigation through complex concepts.

  There are sections where references are indicated but not listed. I would suggest that references be included in these sections as appropriate (see attached file)   I would suggest considering the incorporation of visual aids such as graphs, charts, or figures to complement the textual content of the article. Visual representations have the potential to provide readers with a clearer understanding of complex concepts and findings, thereby enhancing the overall readability and accessibility of your research.   It would be beneficial for the authors to include a section discussing potential future directions for research in this area. Identifying gaps in knowledge and proposing hypotheses or research questions could stimulate further investigation and advancement of the field.

Reviewer 3 Report

Comments and Suggestions for Authors

The author and his team have done an interesting work titled “The role of electroencephalogram-assessed bandwidth power in response to hypnotic analgesia”. while the study was conducted effectively, there are few recommendations for improvement as below. 

Comment1: In the second part, the author first introduced the “Treatment Mediators and Moderators” followed by the third part” Electroencephalography-Assessed Bandwidth Power as a Possible Mediator, Predictor, and Moderator of Hypnosis Treatment”. Furthermore, the methodology for searching PubMed, mentioned multiple times in the third part, raises a question: would a systematic review approach be more appropriate for this study?

Comment 2: To enhance clarity and provide more detail about this aspect of the study, it is recommended that the author included tables of figure summarizing the papers that were included or relevant to the research.

Comment3: Please explain what’s the “(1) aseline bandwidth power as predictors of response to hypnotic analgesia in the Bandwidth Power and Hypnotic Analgesia part. Its baseline or something else, or the wrong typing?

Comment4: The author explores the role of electroencephalogram-assessed bandwidth power in the context to hypnotic analgesia. However, based on my knowledge, different anesthetics and EEG monitors can influence the power readings. Could you please provide further clarification on this matter?

Comment5: Additionally, I noticed that the references cited by the author were predominantly quite old.  Would it be beneficial for the author to include more recent research in their analysis? (García PS, Kreuzer M, Hight D, Sleigh JW. Effects of noxious stimulation on the electroencephalogram during general anaesthesia: a narrative review and approach to analgesic titration. Br J Anaesth. 2021;126(2):445-457. doi:10.1016/j.bja.2020.10.036). This paper elaborately summarized the recent research on the noxious stimulation on the electroencephalogram during GA. The EEG response to painful stimuli suggests that the analgesic effects do not completely block the entry of pain signals into the CNS. Identifying and understanding these EEG changes can help to optimize intraoperative analgesic management, such as adjustment of analgesic drug doses according to EEG changes. Future studies need to evaluate whether analgesia guided according to EEG can reduce postoperative pain and affect other prognostic outcomes, but currently we lack sufficient evidence for this and no large-scale clinical trial confirming whether this approach can bring significant clinical benefit. Overall, future studies need to focus on whether this EEG-guided analgesic approach can actually improve patient prognostic outcomes.

Comments on the Quality of English Language

Minor editing of English language required

Round 2

Reviewer 1 Report

Comments and Suggestions for Authors

We thank the authors for their revision, that we really appreciate> The authors have took in consideration all our suggestions. 

Regarding the flowchart, I let the Editor to decide if this figure should be include or not.

Regarding the EEG figure, the authors could ask a clinician to give them EEG recordings  for each rhythm form healthy volunteers. It's what we have done in the past, and it's really fast to realize for free.

The table is really nice and informative. It really helps to get an easy overview of the main message of the article. It should not be in supplementary file, in my opinion, nut in the main article. But I let the Editor to decide. However, I have few minor comments regarding this table:

- each abbreviation used in the table should be referenced in the legend with the complete name.

-  Study of Freeman : is (p<.10) considered as significant by the authors? it's not usual.

- Study of Jensen 2014: 3 control groups but 4 are reported, please correct

- SInce the title of the 3d column is 'control group', please remove the 'control gps' for each study. It's redundant and not useful.

- Please note that you have a second table 1 in the document, that seems the first draft of the final one. 

Author Response

Response to reviewer

We appreciate the reviewer’s comments and suggestions for improving the the paper further. Here, we  provide a point-by-point description of how we addressed each issue raised.

Reviewer 1

Regarding the flowchart, I let the Editor to decide if this figure should be include or not.

Author response: We are also fine with the Editor's decision on this, although, as noted in our response, we believe that a flow chart is not necessary here.

Regarding the EEG figure, the authors could ask a clinician to give them EEG recordings  for each rhythm form healthy volunteers. It's what we have done in the past, and it's really fast to realize for free.

Author response: It might be possible for us to identify a clinician who could provide us with some images of recordings representing each bandwidth. However, even if we did, someone would need to create a figure based on the images provided. This requires the time of someone who has access to software to create a professional-looking figure and the skill to use this software.  Unfortunately, we do not have the resources for this.

The table is really nice and informative. It really helps to get an easy overview of the main message of the article. It should not be in supplementary file, in my opinion, nut in the main article. But I let the Editor to decide.

Author response: Thank you for your positive words about the we created in response to the reviewers’ suggestions. We agree that the information is useful and improves the paper.  We also agree that it should be in the published article and not be provided as supplemental material, if possible.  We submitted the revision with this in mind. Like the reviewer, however, we defer to the editor to make a final decision on this issue.

However, I have few minor comments regarding this table:

- each abbreviation used in the table should be referenced in the legend with the complete name.

Author response: Thank you for this suggestion. Added a note to Table 1 provide this information.

Study of Freeman : is (p<.10) considered as significant by the authors? it's not usual.

Author response: We agree that a p value of < .10 is unusual. However, Freeman et al. used this p value (their rationale was, due to their limited sample size, using the more standard p < .05 level would result in significant Type II errors), and the goal of the information presented in Table 1 is to summarize the published findings. This requires that we summarize Freeman's results as they were presented. We specifically noted the p value that Freeman et al. used to allow readers to interpret the findings with this knowledge.

- Study of Jensen 2014: 3 control groups but 4 are reported, please correct

Author response: Thank you for noting this error. It was corrected in the revised version of the table.

- SInce the title of the 3d column is 'control group', please remove the 'control gps' for each study. It's redundant and not useful.

Author response: Thank you for noting this issue. We revised the content in this column accordingly.

- Please note that you have a second table 1 in the document, that seems the first draft of the final one.

Author response: Thank you for noting this. We submitted just one version of the Table with this revision.